# Diagnostic Performance of High-Resolution Vessel Wall Magnetic Resonance Imaging and Digital Subtraction Angiography in Intracranial Vertebral Artery Dissection

**DOI:** 10.3390/diagnostics12020432

**Published:** 2022-02-08

**Authors:** Jiwook Ryu, Kyung Mi Lee, Hyug-Gi Kim, Seok Keun Choi, Eui Jong Kim

**Affiliations:** 1Department of Neurosurgery, Kyung Hee University College of Medicine, Kyung Hee University Hospital, Seoul 02447, Korea; drjoshuaryu@gmail.com (J.R.); nscsk@khu.ac.kr (S.K.C.); 2Department of Radiology, Kyung Hee University College of Medicine, Kyung Hee University Hospital, Seoul 02447, Korea; khyukgi@gmail.com (H.-G.K.); euijkim@hanmail.net (E.J.K.)

**Keywords:** vessel wall imaging, digital subtraction angiography, diagnostic performance, vertebral artery, dissection

## Abstract

Purpose: Intracranial vertebral artery dissection (VAD) is being increasingly recognized as a leading cause of Wallenberg syndrome and subarachnoid hemorrhage. Conventional angiography is considered the standard diagnostic modality, but the diagnosis of VAD remains challenging. This study aimed to compare the diagnostic performance of high-resolution vessel wall imaging (HR-VWI) with digital subtraction angiography (DSA) for intracranial VAD. Materials and methods: Twenty-four patients with 27 VADs, who underwent both HR-VWI and DSA within 2 weeks, were consecutively enrolled in the study from March 2016 to September 2020. HR-VWI and DSA were performed to diagnose VAD and to categorize its angiographic features as either definite dissection or suspicious dissection. Features of HR-VWI were used to evaluate direct arterial wall imaging. The reference standard was set from the clinicoradiologic diagnosis. Two independent raters evaluated the angiographic features, dissection signs, and interrater agreement. Each subject was also dichotomized into two groups (suspicious or definite VAD) in each modality, and diagnosis from HR-VWI and DSA was compared with the final diagnosis by consensus. Results: HR-VWI had higher agreement (90.6% vs. 53.1%) with the final diagnosis and better interrater reliability (kappa value (κ) = 0.91; 95% confidence interval (CI) = 0.64–1.00) compared with DSA (κ = 0.58; 95% CI = 0.35–1.00). HR-VWI provided a more detailed identification of dissection signs (77.7% vs. 22.2%) and better reliability (κ = 0.88; 95% CI = 0.58–1.00 vs. κ = 0.75; 95% CI = 0.36–1.00), compared to DSA. HR-VWI was comparable to DSA for the depiction of angiographic features for VAD. Conclusions: HR-VWI may be useful to evaluate VAD, with better diagnostic confidence compared to DSA.

## 1. Introduction

Intracranial artery dissection (IAD) refers to hemorrhage within the wall of an intracranial artery, regardless of the etiology (e.g., spontaneous or traumatic) [1]. IAD is considered an uncommon and underdiagnosed cause of stroke [2], accounting for approximately 2% of ischemic strokes and 5–25% of strokes in young and middle-aged patients [3,4]. IAD causes arterial stenosis, occlusion, and aneurysm, which in turn lead to cerebral ischemia, subarachnoid hemorrhage (SAH), isolated headache, and symptoms associated with mass effect [2,5]. IAD occurs predominantly in the posterior circulation in East Asian countries and in the anterior circulation in Western countries [6]. Intracranial vertebral artery dissection (VAD) is a component of IAD (Figure 1) and has become increasingly recognized as one of the main causes of Wallenberg syndrome and SAH [7,8,9].

Early diagnosis of VAD is often challenging due to the small size of the intracranial artery [2]. Although catheter-based digital subtraction angiography (DSA) has been the gold standard for the diagnosis and evaluation of VAD, the diagnostic accuracy of three-dimensional (3D) magnetic resonance (MR) images has rapidly developed in recent years [10]. This technique not only helps to delineate the shapes of the arteries, but also assess blood flow in the vessels. In contrast, DSA only reveals geometric shapes of the vessel lumen and does not allow direct configuration of the arterial wall. Furthermore, VAD can present with a fusiform dilatation, segmental stenosis, or occlusion, with the distribution of these angioarchitectures varying widely between studies [11,12]. If only angiography patterns are emphasized, the correct diagnosis is sometimes confused. From a therapeutic point of view, DSA has the advantage that intervention procedures such as stent insertion or thrombectomy can be performed immediately after angiography [13,14]. However, in the case of VAD, many patients receive medical treatment at the time of diagnosis.

Intracranial 3D high-resolution vessel wall imaging (HR-VWI) has recently emerged as a popular modality with clinical applicability [15]. Three-dimensional (3D) HR-VWI enables a resolution of less than 1 mm while allowing various MR sequences, including T1-weighted imaging (T1WI), T2-weighted imaging (T2WI), proton density (PD) imaging, contrast-enhanced T1WI, susceptibility-weighted imaging, and quantitative susceptibility mapping [16,17]. This technique has played an important role in the assessment of various cerebrovascular diseases, such as intracranial atherosclerotic disease, dissection, Moyamoya disease, vasculitis, and reversible cerebral vasoconstriction syndrome [18]. It also offers excellent visualization of both the arterial wall and lumen to confirm the radiological features of IAD [19,20,21,22,23,24]. A recent study demonstrated that HR-VWI is a useful and reliable diagnostic tool for the diagnosis of VAD [25].

To the best of our knowledge, no previous study has directly compared HR-VWI and DSA findings in VAD. Considering that VAD is a consequence of an underlying “vessel wall disease”, we hypothesized that HR-VWI and DSA may have differences in the diagnostic performance for VAD. This study compared the diagnostic reliability and accuracy of HR-VWI and DSA, based upon their radiological findings for VAD.

## 2. Materials and Methods

### 2.1. Patient Selection

This retrospective study was approved by our institutional ethical committees, and the need for informed consent was waived. A total of 532 patients underwent HR-VWI between March 2016 and September 2020. Among them, 500 patients were excluded because they were diagnosed with intracranial atherosclerotic disease (343 patients), cerebral aneurysm (125 patients), Moyamoya disease (19 patients), and undetermined causes (13 patients). Thirty-two patients were diagnosed with IADs, which were located at the V4 segment of the vertebral artery (VA, 24 patients), the internal carotid artery (ICA, 4 patients), the middle cerebral artery (MCA, 3 patients), and the anterior cerebral artery (ACA, 1 patient). Eventually, 24 patients (13 males and 11 females, mean age = 56.6 years, age range = 31–78 years) with 27 VADs were enrolled in this study. Three patients had both side VADs. Details of the flowchart of the study population are shown in Figure 2.

Patients were retrospectively enrolled in accordance with the following criteria: (1) diagnosed with VAD based on clinical and radiological findings; (2) more than two imaging modalities (essentially both HR-VWI and DSA) were used to diagnose VAD; (3) no history of head trauma; (4) underwent HR-VWI within 7 days of symptom onset; and (5) underwent DSA within 2 weeks of HR-VWI. Demographic data including sex, age, risk factors associated with acute stroke (i.e., hypertension, diabetes, hyperlipidemia), clinical symptoms, and treatment options (i.e., medical treatment or surgical treatment) were obtained from the Picture archiving and communication system (PACS) and electronic medical records.

### 2.2. Imaging Protocol

#### 2.2.1. Magnetic Resonance Image (MRI) Protocol

Three-dimensional (3D) HR-VWI was performed using an Achieva 3.0-Tesla scanner with a 64-channel head coil (Philips Medical Systems, Best, the Netherlands) and a VIDA 3.0-Tesla scanner with a 128-channel head and neck coil (Siemens, Erlangen, Germany). Three-dimensional (3D) sequences were acquired using 3D turbo spin echo, with variable flip angle refocusing radiofrequency pulses. The VWI protocol included intracranial time-of-flight magnetic resonance angiography (TOF-MRA), maximum-intensity projection (MIP), T1WI, PD image, and contrast-enhanced T1WI. The following sequences were applied: in Achieva, T1W spin echo (in-plane saturation, 0.5 × 0.5 mmL; slice thickness, 0.5 mm; repetition time (TR)/echo time (TE), 600/17 ms; field of view, 160 × 160; time, 7 min 20 s), PD image (in-plane saturation, 0.5 × 0.5 mmL; slice thickness, 0.5 mm; TR/TE, 600/17 ms; field of view, 160 × 160; time, 7 min 20 s), and contrast-enhanced T1WI after intravenous gadolinium administration (Gadovist; Bayer Schering Pharma, Berlin, Germany). The total acquisition time was approximately 30 min. In the VIDA vendor, the MR sequences were as follows: T1W spin echo (in-plane saturation, 0.5 × 0.5 mmL; slice thickness, 0.5 mm; TR/TE, 700/13 ms; field of view, 180 × 180; time, 7 min 5 s), PD image (in-plane saturation, 0.5 × 0.5 mmL; slice thickness, 0.5 mm; TR/TE, 1200/23 ms; field of view, 180 × 180; time, 7 min 5 s), and contrast-enhanced T1WI after intravenous gadolinium administration (Prohance; Bracco, Milan, Italy). The total acquisition time was approximately 22 min. The 3D images were reconstructed into coronal, sagittal, and axial images and displayed with an isovoxel of 0.5 × 0.5 × 0.5 mm^3^.

#### 2.2.2. DSA Imaging Protocol

DSA was performed on the Philips Allura Xper FD angiography suite (Philips, the Netherlands) with a biplane system. Seldinger’s technique was used to obtain patients’ images of the ICA and both VA, including the anterior–posterior and lateral projections. Routine angiography acquisitions are typically performed at 4 F/s for the first 3 s, 2 F/s for the next 3 s, and 1 F/s thereafter, while fluoroscopy sequences involve 15 F/s. The rotational 3D volumetric sequence was optimized using a standard protocol as follows: operating at 30 F/s, 84 kVp, 210 mAs, and 122 images.

The transfemoral cerebral angiography (TFCA) protocols of our institute are as follows: (1) The anterior wall of the femoral artery is punctured at the level of the bottom of the femoral head with the Seldinger technique. (2) Five Fr catheters are used in adult patients, and the catheterization is performed as gently as possible. (3) Selective catheterization of both carotid arteries (common, internal, external) and VA (routinely the left side) is usually performed, and road map images are taken at each step of the selective catheterizations. (4) A 100 cm Bentson (JB2) diagnostic catheter (Cordis) is initially used in adult patients >40 years old, and a 100 cm Bern catheter is used in adults <40 years old. If this fails, a 100 cm Simon2 catheter (Cook Medical) is used in sequence depending on the complexity of the aortic arch and the configuration of its major branches. Catheters are positioned as proximally as possible. (5) Intracranial angiography is performed on the ICA. (6) The operator should be fully aware of the constant dye (Visipaque 270, GE Healthcare) concentration (100 mL of dye mixed with 40 mL of N/S), injection speed, and volume (4 mL/s and 6 mL volume in the ICA; 5 mL/s and 7 mL volume in the VA). (7) The femoral sheath is removed immediately after completion of angiography, and manual compression is applied for 20 min to achieve hemostasis.

### 2.3. Image Analysis

MRI was independently reviewed by a neuroradiologist (K.M.L. with 10 years of experience in the neuroradiologic field) and a neurointerventionalist (J.R. with 7 years of experience in vascular neurosurgery and the neurointervention field). Both reviewers were blinded to the patients’ clinical information and angiographic information. The reviewers focused on the abnormal features on intracranial TOF-MRA, including the MIP image and MRA source image, before performing an analysis focusing on HR-VWI as the following findings: (1) fusiform dilation, (2) pearl–string sign or string sign, (3) intimal flap or double-lumen sign, and (4) intramural hematoma (IMH) or dissecting aneurysm [23,24,26] (Figure 3).

The radiologic findings of DSA were also independently reviewed by the same two raters, both of whom were blinded to clinical information. The radiologic features included (1) fusiform dilation, (2) pearl–string sign or string sign, and (3) intimal flap or double-lumen sign [19,26].

After imaging analysis, each case was dichotomized as a “definite” or “suspicious” VAD in each modality and was classified according to the diagnostic criteria of the Strategies against Stroke Study for Young Adults in Japan [27] (Table 1).

One month after initial analysis in each imaging modality, two neuroradiologists (K.M.L. and J.R.) and a neurointerventionalist (E.J.K. with 20 years of experience in neuroradiology and the neurointervention field) arrived at the final diagnosis by consensus after reviewing all clinical data and investigations from hospital admission to the last follow up. It was based on the same criteria used in imaging analysis. Any discordance between the three raters was resolved by consensus. Each diagnosis based on HR-VWI and DSA was compared to the final diagnosis.

### 2.4. Statistical Analysis

All findings were dichotomized, and the prevalence of each finding was calculated for each observer. To compare the diagnostic accuracy and sensitivity between 3D HR-VWI and DSA, a McNemar test and logistic regression using a generalized estimating equation (GEE) that accounted for the clustering of the same patient were performed. The comparison of the positive predictive value (PPV) was tested using logistic regression utilizing a GEE. A value of *p* < 0.05 was considered to indicate a significant difference.

The interobserver agreement for each radiological finding and VAD assessment across the two modalities were determined by using Cohen’s kappa statistics. The strength of agreement of the κ values was categorized as follows: <0.20, poor; 0.21–0.40, fair; 0.41–0.60, moderate; 0.61–0.80, good; and 0.81–1.00, excellent [28].

Receiver operating characteristic (ROC) curves were generated to determine the diagnostic value of each test for the assessing VAD. The area under the curve (AUC) and 95% confidence interval (CI) were also calculated. The discriminatory value of curves was interpreted as excellent (0.9 to 1), good (0.8 to 0.89), fair (0.7 to 0.79), poor (0.6 to 0.69), or as failing or having no discriminatory capacity (0.5 to 0.59) [29,30]. All the statistical analyses were performed using SPSS 24.0 software (SPSS Inc., Chicago, IL, USA).

## 3. Results

There were 13 men and 11 women patients, with a mean age of 56.6 years (range = 31–78). A total of 27 intracranial vertebral arteries (V4 segment) were enrolled in the study. The main risk factors for VAD included hypertension (14 of 24, 58.3%), hyperlipidemia (11 of 24, 45.8%), and diabetes mellitus (6 of 24, 25%). Clinical presentations related to VAD included headache (11 of 24, 45.8%), ischemic stroke (6 of 24, 24%), dizziness (4 of 24, 16.6%), and SAH (3 of 24, 12.5%). Table 2 summarizes the patients’ demographic data.

### 3.1. Diagnostic Performance of HR-VWI Compared with MRA and DSA

HR-VWI had higher agreement with the final diagnosis than DSA (Table 3). On HR-VWI images, the prevalence of definite VAD was 19 of 27 VADs (70.3%), and the interobserver agreement was excellent (κ = 0.91; 95% CI = 0.67–1.00). The raters corroborated the final diagnosis in 25 of the 27 VADs (92.5%). For DSA, the prevalence of definite VAD was 7 of 27 VADs (25.9%), which resulted in moderate agreement (κ = 0.58; 95% CI = 0.13–0.89).

### 3.2. Radiological Performance of HR-VWI Compared with MRA and DSA

HR-VWI was comparable to DSA for the depiction of angiographic features for VAD (Table 4). Fusiform dilatation was revealed in 12 of 27 VADs (44.4%) and 11 of 27 VADs (40.7%) on HR-VWI and DSA, respectively. The pearl–string sign was observed in 16 of 27 VADs (59.2%) in both modalities. The interobserver agreement of fusiform dilatation and pearl–string sign was good (κ = 0.78; 95% CI = 0.55–1.00 and κ = 0.76; 95% CI = 0.49–1.00) for HR-VWI and good (κ = 0.77; 95% CI = 0.49–1.00 and κ = 0.69; 95% CI = 0.37–0.93) for DSA. In contrast, HR-VWI provided better reliability and a more detailed identification of dissection sign than DSA. The detection rate of double lumen and intimal flap sign was 77.7% (21 of 27 VADs) and 22.2% (6 of 27 VADs), respectively, and the interobserver agreement was excellent (κ = 0.88; 95% CI = 0.58–1.00) and good (κ = 0.75; 95% CI = 0.36–1.00) in HR-VWI and DSA, respectively. Intramural hematoma was visualized in 20 of 27 VADs (74.0%) only in HR-VWI, and the interobserver agreement was good (κ = 0.72; 95% CI = 0.36–1.00).

### 3.3. Comparison of the Diagnostic Accuracy, Sensitivity, and PPV between HR-VWI with MRA and DSA

Compared to DSA, HR-VWI showed overall better diagnostic accuracy for diagnosing VAD (91.9 of HR-VWI and 85.5 of DSA, *p* < 0.03). The sensitivity of HR-VWI was significantly higher than that of DSA (92.4 of HR-VWI and 79.1 of DSA, *p* < 0.05). The PPVs of HR-VWI were also significantly higher than those of DSA (100 of HR-VWI and 73 of DSA, *p*-value is not applicable for calculation). The final diagnosis was reached by consensus of the three readers after reviewing all clinical and radiological data.

### 3.4. ROC Curves

The ROC curves for HR-VWI and MRA and DSA are shown in Figure 4. The AUC for HR-VWI was 0.95 (95% CI = 0.87–1.00), indicating that HR-VWI demonstrated excellent diagnostic performance in the identification of definite VAD. In contrast, DSA had an AUC of 0.67 (95% CI = 0.45–0.88), indicating that it had poor diagnostic value.

## 4. Discussion

DSA is widely considered the gold standard for diagnosing intracranial artery pathologies [31], with advantages including accuracy in assessing various geometric shapes of the affected vessel, relevant hemodynamic information, and assessment of collateral circulation. DSA also offers superior spatial resolution compared to other methods of angiography, such as MRA and computed tomography angiography (CTA) [32,33]. Furthermore, neurosurgeons and neurointerventionalists have used DSA for both diagnostic and therapeutic purposes. In extracranial cervical artery dissection cases, DSA findings are helpful to diagnose dissection compared to MRA. However, in VAD cases, DSA without the assistance of MR findings is not helpful in the diagnosis of dissection because it does not reveal the imaging features of the arterial wall, as proven in this study. The angiographic appearance of VAD is nonspecific because other causes such as thromboembolism or atherosclerotic disease may present with similar angiographic characteristics. HR-VWI has been proven to have the ability to look beyond the vessel lumen directly and is the only reliable possibility to diagnose VAD (Figure 5). In this study, we demonstrated the diagnostic performance of HR-VWI and DSA independently. To the best of our knowledge, this is the first study to directly compare HR-VWI and DSA findings in VAD. The results suggest that HR-VWI is superior to DSA for diagnosing VAD. HR-VWI provided a more detailed identification of dissection signs (77.7% vs. 22.2%) and had better reliability (κ = 0.88; 95% CI = 0.58–1.00 vs. κ = 0.75; 95% CI = 0.36–1.00), compared to DSA.

The pathomechanism of VAD still remains unclear, although intimal tear is the generally accepted mechanism for VAD development. Some studies have demonstrated a disruption of the internal elastic lamina and the media of the arterial wall in surgical specimens [34,35]. However, it is very difficult to detect intimal tears at the time of microscopic examination. The lack of communication between the true and false lumen indicates that some cases may result from a primary IMH [3]. The IMH, i.e., false lumen, is located within the layers of the tunica media, but it may be eccentric, either toward the intima or the adventitia [36]. A subintimal dissection tends to result in arterial stenosis, whereas a subadventitial dissection can cause aneurysmal dilatation [36]. Ono et al. showed that the intramural hemorrhage was replaced by granulation tissue 2 weeks after onset, followed by compensatory intimal thickening [34]. Next, neovascularization was observed in the thickened intima 4 weeks from symptom onset, resulting in the formation of a fusiform aneurysm [34].

The imaging techniques for the diagnosis of VAD include conventional angiography, CT, and MRI. The standard in diagnostic imaging is still represented by DSA [37]. The pathognomonic radiological findings indicating dissections include: intimal flap, double lumen, intramural hematoma, aneurysmal dilatation, tapered stenosis, and occlusion. Han et al. showed the intimal flap sign was the most common finding and was identified on MRI in more than 90% of VAD patients [25], while IMH was observed in more than 50% of these patients [25]. IMH usually leads to a regular crescent-shaped thickening of the arterial wall with enlargement of the external diameter of the dissected artery and often a reduced and eccentric arterial lumen [38]. Fusiform dilations were more common in VAD with SAH compared to VAD without SAH [11]. Both segmental stenosis and occlusion in SAH are highly suggestive of VAD. However, in VAD without SAH, these findings are not pathognomonic (specifically characteristic or indicative of a particular disease). Likewise, a fusiform dilation located at a nonbranching site is very suggestive of VAD if associated with segmental stenosis, but it is not specific for isolated VAD [39]. Additional radiological investigations are warranted to confirm the diagnosis of VAD.

HR-VWI showed a comparable to better diagnostic accuracy and confidence value than DSA in diagnosing VAD. The diagnostic pearls of dissection are the intimal flap (with double lumen), followed by IMH (with an eccentric flow void of the patent lumen) and aneurysmal dilation with stenosis (string of pearl sign). These findings are well visualized by HR-VWI but not DSA; thus, HR-VWI can provide better spatial resolution to evaluate vessel wall pathology. The presence of contrast enhancement has been used to evaluate plaque components and the degree of neovascularization [40]. Perivascular enhancement may be another indicator of the acuteness of the dissection and suggests an inflammatory component of VAD [20]. Moreover, a previous study revealed that more VADs in patients with acute ischemic stroke showed more enhancement than VAD in patients without acute ischemic stroke [41]. Although it is for research purposes, to date, hemodynamic study is possible using HR-VWI.

The excellent performance of HR-VWI in the diagnosis of VAD originated from the higher detection rate for dissection signs. Better assessment of dissection signs is of particular importance for the diagnosis of VAD. The aforementioned criteria demonstrated that the double-lumen and intimal flap sign were the main findings indicating “definite” dissection [27]. A previous study also showed that the dissection flap sign had the greatest diagnostic value, and HR-VWI can be a useful modality for VAD [25]. In contrast, a DSA study showed that signs indicating dissection are observed in only 10% of IAD cases.

Our study has some limitations. First, this was a retrospective study and enrolled only a few patients, which may result in limitations in its statistical significance and selection bias. Second, the diagnoses in our study may be insufficient because they were made according to the aforementioned diagnostic criteria, and the conclusive diagnosis with pathologic confirmation in VAD was not acquired. IAD was usually diagnosed based on clinical, laboratory, and radiological information without pathologic confirmation, and physicians sometimes must diagnose challenging cases in the clinical field [42]. The enrolled patients also presented challenging cases for both DSA and HR-MR, which may introduce a selection bias and result in an imbalance of the analyzed arterial segments in some cases. Furthermore, we classified the results as binary (“0” or “1”) due to the small size of the data set. The ROC curve showed a typical curve pattern because there was only one point between (0,0) and (1,1). Despite these limitations, we believe that our study substantially reflects actual clinical practice.

This study is the first to compare HR-VWI and DSA findings in VAD numerically and statistically. HR-VWI is an excellent imaging technique for the evaluation of VAD and provides superior performance in the assessment of VAD compared with DSA. The main difference between HR-VWI and DSA lies in the detection of the dissection signs. Thus, HR-VWI can be used as a standard modality for the assessment of VAD in initial and follow-up investigations.

## Figures and Tables

**Figure 1 diagnostics-12-00432-f001:**
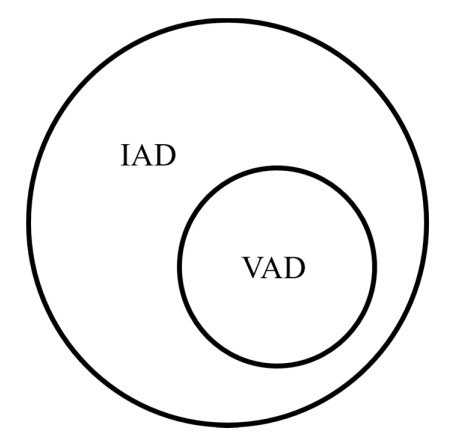
Diagram showing the relationship between intracranial artery dissection (IAD) and vertebral artery dissection (VAD).

**Figure 2 diagnostics-12-00432-f002:**
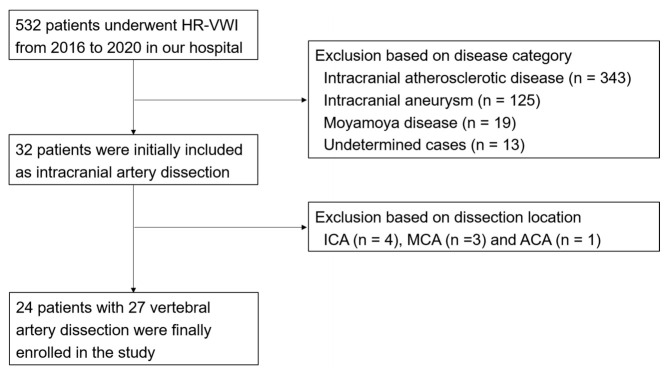
Flow diagram of the patient cohorts for assessment of intracranial vertebral artery dissection. Abbreviations: HR-VWI, high-resolution vessel wall imaging; ICA, internal carotid artery; MCA, middle cerebral artery; ACA, anterior cerebral artery.

**Figure 3 diagnostics-12-00432-f003:**
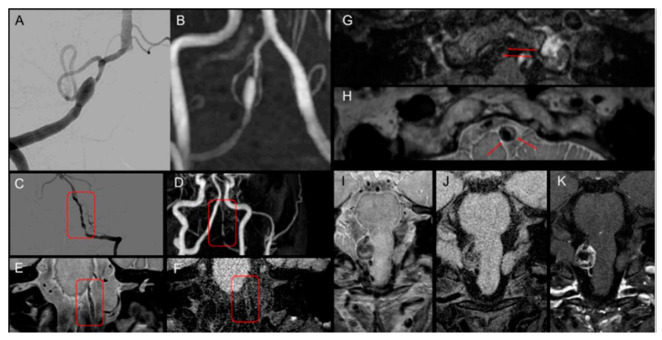
Radiographic features of vertebral artery dissection. (**A**,**B**) Fusiform dilatation in DSA and MRA. (**C**–**F**) Stenosis with dilatation called a string of pearls signs in DSA and HR-VWI. (**G**,**H**) Double-lumen sign in HR-VWI. (**I**–**K**) Dissecting aneurysm in HR-VWI. Abbreviations: DSA, digital subtraction angiography; MRA, magnetic resonance angiography; HR-VWI, high-resolution vessel wall imaging.

**Figure 4 diagnostics-12-00432-f004:**
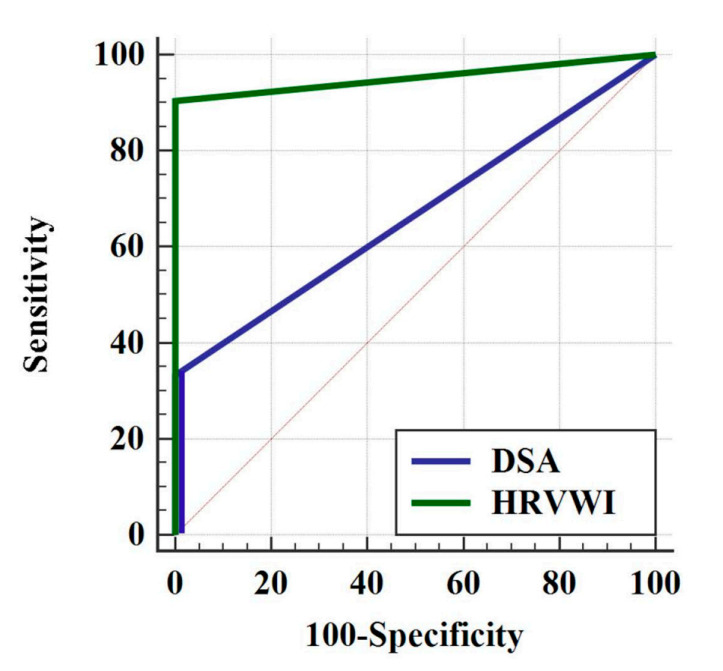
ROC curve for high-resolution vessel wall imaging (HR-VWI) and digital subtraction angiography (DSA). The area under the receiver operating characteristic curve is 0.95 (95% CI = 0.87–1.00) for HR-VWI and 0.67 (95% CI = 0.45–0.88) for DSA. (CI; confidence interval).

**Figure 5 diagnostics-12-00432-f005:**
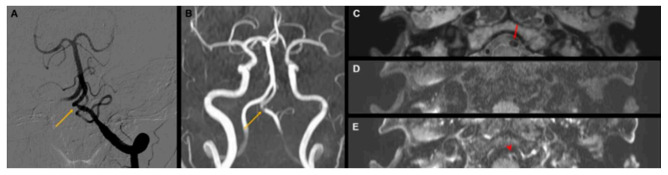
A representative case of vertebral artery dissection. A 57-year-old woman patient presented with severe headache. Cerebral angiography (**A**) and time-of-flight MRA (**B**) showing the only focal stenosis at the left V4 segment of the vertebral artery (yellow arrow). (**C**) The intimal flap sign (red arrow) presents as iso-signal intensity on proton-density high-resolution vessel wall imaging (HR-VWI). The double-lumen sign is also observed. (**D**,**E**) Periarterial enhancement (red arrowhead) on contrast-enhanced T1-weighted HR-VWI.

**Table 1 diagnostics-12-00432-t001:** Diagnostic criteria for intracranial artery dissection.

Diagnostic Criteria
*Definite case*
Satisfying any of Diagnostic Criteria I, II, or III below
I. Either intimal flap or double lumen visible on cerebral angiogram
II. Either intimal flap or double lumen visible on MRI or MRA (tomogram image). Handled identically if the transverse image on 3D-CTA and ultrasound examination is sufficiently delineated and a clear intimal flap and double lumen are visible.
III. If any of Findings IV, V, or VI are observed and a clear change is seen in the findings over time with repeated imaging examinations. Limited to cases in which a cause other than dissection can be ruled out
*Suspected case*
Satisfying any of Diagnostic Criteria IV, V, or VI below
IV. Nonspecific findings suggesting arterial dissection (pearl sign, tapered occlusion) are visible on cerebral angiogram other than the findings in I above
V. Findings are visible on MRA angiogram that appear to correspond to the pearl and string sign, string sign, or tapered occlusion on cerebral angiogram
VI. Intense signal suggesting intramural hematoma visible on MRI T1-weighted image

Referenced by the Strategies against Stroke Study for Young Adults in Japan criteria [27]. Abbreviations: MRI, magnetic resonance imaging; MRA, magnetic resonance angiography; CTA, computed tomography angiography.

**Table 2 diagnostics-12-00432-t002:** Clinical characteristics of study population.

Demographics	
**No. of patients (lesions)**	24 (27)
No. of female (%)	11 (45.8%)
Mean age (range)	56.6 (31–78)
**Risk factors**	
Hypertension	14 (58.3%)
Hyperlipidemia	11 (45.8%)
Diabetes mellitus	6 (25.0%)
**Clinical Presentation**	
Headache	11 (45.8%)
Ischemic stroke	6 (24%)
Dizziness	4 (16.6%)
Subarachnoid hemorrhage	3 (12.5%)
**Treatment**	
Medication	16 (66.6%)
Intervention	8 (33.4%)

**Table 3 diagnostics-12-00432-t003:** Diagnostic performance of HR-VWI and DSA compared to final diagnosis.

	Final Diagnosis		Total
	Definite VAD	Suspicious VAD	
HR-VWI			
Definite VAD	19	0	19
Suspicious VAD	2	6	8
Total	21	6	27
DSA			
Definite VAD	7	0	7
Suspicious VAD	14	6	20
Total	21	6	27

Abbreviations: DSA, digital subtraction angiography; HR-VWI, high-resolution vessel wall imaging; VAD; vertebral artery dissection.

**Table 4 diagnostics-12-00432-t004:** Comparison of radiological features and performance of vertebral artery dissection between high-resolution vessel wall imaging with MR angiography and digital subtraction angiography.

	HR-VWI	DSA
**Angiographic finding**		
Fusiform dilatation	12 (44.4%)	11 (40.7%)
Interobserver agreement	0.78 (95% CI = 0.55–1.00)	0.77 (95% CI = 0.49–1.00)
Pearl–string sign	16 (59.2%)	16 (59.2%)
Interobserver agreement	0.76 (95% CI = 0.49–1.00)	0.69 (95% CI = 0.37–0.93)
**Dissection sign**		
Double-lumen sign or intimal flap	21 (77.7%)	6 (22.2%)
Interobserver agreement	0.88 (95% CI = 0.58–1.00)	0.75 (0.36–1.00)
Intramural hematoma	20 (74.0%)	Not evaluated
Interobserver agreement	0.72 (95% CI = 0.36–1.00)	
**Diagnosis**		
Definite VAD	19 (70.3%)	7 (25.9%)
Interobserver agreement	0.91 (95% CI = 0.67–100)	0.58 (95% CI = 0.13–0.89)

Abbreviations: DSA, digital subtraction angiography; HR-VWI, high-resolution vessel wall imaging; VAD, vertebral artery dissection, CI; confidence interval.

## Data Availability

Not applicable.

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
