# Peer review of "Diagnostic Performance of High-Resolution Vessel Wall Magnetic Resonance Imaging and Digital Subtraction Angiography in Intracranial Vertebral Artery Dissection"

_diagnostics, 2022, doi:10.3390/diagnostics12020432_

Round 1

Reviewer 1 Report

The manuscript has been improved.

Author Response

The manuscript was carefully revised by a native English speaker again.

Reviewer 2 Report

Interesting and well written study regarding the diagnostic performance of high-resolution vessel wall magnetic resonance imaging and digital subtraction angiography in intracranial vertebral artery dissection.

In general, the manuscript is interesting, the investigation is acceptable, the paper is clear and well presented. However, there are some suggestions that the authors should try to apply as much as possible in order to correct some details and also improve the overall level of the manuscript to merit publication in this journal, here are a few comments:

- I would add some lines in introduction regarding the endovascular treatment of a single vertebral artery occlusion, since this became evident with some recent studies [ Siebert E, et al. Revascularization Techniques for Acute Basilar Artery Occlusion : Technical Considerations and Outcome in the Setting of Severe Posterior Circulation Steno-Occlusive Disease. Clin Neuroradiol. 2019 Sep;29(3):435-443. doi: 10.1007/s00062-018-0683-3. Epub 2018 Apr 12. PMID: 29651586. Alexandre AM. Posterior Circulation Endovascular Thrombectomy for Large-Vessel Occlusion: Predictors of Favorable Clinical Outcome and Analysis of First-Pass Effect. AJNR Am J Neuroradiol. 2021 May;42(5):896-903. doi: 10.3174/ajnr.A7023. Epub 2021 Mar 4. PMID: 33664106; PMCID: PMC8115369. ]

- Conclusion is not supported by datas. Nowadays DSA remains the best diagnostic tool for vessel imaging. I would rather say that the hematoma can be better visualised with MRI compared to DSA, and above all for radiologist with no experience with DSA is probably simpler to evaluate a dissection with the MRI.

Author Response

1.I would add some lines in introduction regarding the endovascular treatment of a single vertebral artery occlusion, since this became evident with some recent studies [ Siebert E, et al. Revascularization Techniques for Acute Basilar Artery Occlusion : Technical Considerations and Outcome in the Setting of Severe Posterior Circulation Steno-Occlusive Disease. Clin Neuroradiol. 2019 Sep;29(3):435-443. doi: 10.1007/s00062-018-0683-3. Epub 2018 Apr 12. PMID: 29651586. Alexandre AM. Posterior Circulation Endovascular Thrombectomy for Large-Vessel Occlusion: Predictors of Favorable Clinical Outcome and Analysis of First-Pass Effect. AJNR Am J Neuroradiol. 2021 May;42(5):896-903. doi: 10.3174/ajnr.A7023. Epub 2021 Mar 4. PMID: 33664106; PMCID: PMC8115369. ]

==> I appreciate your suggestions. “Introduction” section was revised based on your instructions. We add some explanations regarding the endovascular treatment as follows:

From a therapeutic point of view, DSA has the advantage that intervention procedure such as stent insertion or thrombectomy can be performed immediately after angiography [13, 14]. However, in the case of VAD, many of patients receive the medial treatment at the time of diagnosis.

Ref 13. Siebert E, et al. Revascularization Techniques for Acute Basilar Artery Occlusion : Technical Considerations and Outcome in the Setting of Severe Posterior Circulation Steno-Occlusive Disease. Clin Neuroradiol. 2019 Sep;29(3):435-443. doi: 10.1007/s00062-018-0683-3. Epub 2018 Apr 12. PMID: 29651586. 

Ref 14. Alexandre AM. Posterior Circulation Endovascular Thrombectomy for Large-Vessel Occlusion: Predictors of Favorable Clinical Outcome and Analysis of First-Pass Effect. AJNR Am J Neuroradiol. 2021 May;42(5):896-903. doi: 10.3174/ajnr.A7023. Epub 2021 Mar 4. PMID: 33664106; PMCID: PMC8115369. ]

  1. Conclusion is not supported by datas. Nowadays DSA remains the best diagnostic tool for vessel imaging. I would rather say that the hematoma can be better visualised with MRI compared to DSA, and above all for radiologist with no experience with DSA is probably simpler to evaluate a dissection with the MRI.

==> In our hospital, DSA was unconditionally performed when a patient was suspected of dissection. The results of this manuscript were for VAD patients. Nevertheless, the HR-VWI had to be performed again regardless of the DSA result. For DSA, the prevalence of definite VAD was 7 of 27 VADs (25.9%), interobserver agreement was moderate (κ=0.58; 95% CI, 0.13-0.89). Also, DSA has an AUC of 0.67, indicating with poor diagnostic value. In other words, when stenosis or string of pearls sign was seen on DSA, many patients were not sure if it was a dissection or not. Because of other causes such as thromboembolism or atherosclerotic disease may present very similar angiographic characteristics of VAD.

Of course, in extracranial cervical artery dissection cases, DSA findings are helpful to diagnose the dissection as compared with MRA. However, in VAD cases, DSA is not superior in the diagnosis of dissection without the assistant of MR findings because it doesn’t reveal the imaging features of the arterial wall which were proved in this study.

To our knowledge, this is the first study to directly compare HR-VWI and DSA findings in VAD. This means that this study is meaningful to show the results of HR-VWI and DSA numerically and statistically as follows: HR-VWI provided more detailed identification of dissection sign (77.7% vs. 22.2%) and better reliability (κ=0.88; 95% CI, 0.58-1.00 vs. κ=0.75; 95% CI, 0.36-1.00), compared with DSA.

==> Thank you very much of your comment. We revised conclusion.
